# Impact of the Deletion of Genes of the Nitrogen Metabolism on Triacylglycerol, Cardiolipin and Actinorhodin Biosynthesis in *Streptomyces coelicolor*

**DOI:** 10.3390/microorganisms12081560

**Published:** 2024-07-30

**Authors:** Sonia Abreu, Clara Lejeune, Michelle David, Pierre Chaminade, Marie-Joelle Virolle

**Affiliations:** 1Lip (Sys)2 (Lipides Systèmes Analytiques et Biologiques), UFR Pharmacie-Bâtiment Henri Moissan, CNRS, CEA, Université Paris-Saclay, 17 Avenue des Sciences, 91400 Orsay, France; sonia.abreu@universite-paris-saclay.fr (S.A.); pierre.chaminade@universite-paris-saclay.fr (P.C.); 2Institute for Integrative Biology of the Cell (I2BC), Department of Microbiology, Group “Energetic Metabolism of Streptomyces”, CNRS, CEA, Université Paris-Saclay, 1 Avenue de la Terrasse, 91198 Gif-Sur-Yvette, Francemichelle.david@i2bc.paris-saclay.fr (M.D.)

**Keywords:** nitrogen limitation, triacylglycerol, cardiolipin, oxidative stress, antibiotic

## Abstract

Since nitrogen limitation is known to be an important trigger of triacylglycerol (TAG) accumulation in most microorganisms, we first assessed the global lipid content of 21 strains derived from *Streptomyces coelicolor* M145 deleted for genes involved in nitrogen metabolism. Seven of these strains deleted for genes encoding proteins involved in polyamine (GlnA2/SCO2241, GlnA3/SCO6962, GlnA4/SCO1613), or protein (Pup/SCO1646) degradation, in the regulation of nitrogen metabolism (GlnE/SCO2234 and GlnK/SCO5584), or the global regulator DasR/SCO5231 that controls negatively the degradation of N-acetylglucosamine, a constituent of peptidoglycan, had a higher TAG content than the original strain, whereas five of these strains (except the *glnA2* and *pup* mutants) had a lower cardiolipin (CL) content. The production of the blue polyketide actinorhodin (ACT) was totally abolished in the *dasR* mutant in both Pi conditions, whereas the deletion of *pup*, *glnA2*, *glnA3*, and *glnA4* was correlated with a significant increase in total ACT production, but mainly in Pi limitation. Unexpectedly, ACT production was strongly reduced in the *glnA3* mutant in Pi proficiency. Altogether, our data suggest that high TAG and ACT biosynthesis and low CL biosynthesis might all contribute to the lowering of oxidative stress resulting from nitrogen limitation or from other causes.

## 1. Introduction

Nitrogen (N) and phosphate (P) are constitutive elements of all living cells since, with carbon (C), oxygen (O), hydrogen (H), and sulfur (S), these elements are necessary for the biosynthesis of building blocks (nucleotides and amino acids) used for the biosynthesis of cellular components such as nucleic acids, proteins, membranous phospholipids, cell wall peptidoglycan, etc. However, the inorganic forms of N (ammonium, nitrates, nitrites, etc.) and P are often scarce in the environment of most microorganisms, and these elements are thus usually present in biological molecules such as nucleic acids, proteins, polyamines, lipids, and the cell wall resulting from plant or animal death. These macromolecules have to be extracellularly degraded into smaller compounds to be transported and assimilated by microorganisms. Alternatively, if external inorganic or organic N and P sources are scarce, the bacteria activates the degradation of its own biological molecules in order to re-cycle N and P present in the latter’s.

In this study, we assessed the content of lipids/triacylglycerol (TAG) and the level of production of the specialized metabolite actinorhodin (ACT) of various strains derived from *S. coelicolor* (*SC*) M145 and grown in the classical medium R2YE. These strains were deleted for genes encoding proteins involved in the degradation of proteins (Pup/SCO1646) [1,2] or polyamines (GlnA2/SCO2241, GlnA3/SCO6962, and GlnA4/SCO1613) [3,4,5,6,7] as well as others genes involved in nitrogen metabolism (*glnA/sco2198*, *glnII/sco2210*, and *nnar/sco2958*, *gdhA/sco4683*) or its regulation (*glnR/sco4159*, *glnRII/sco2213*, *glnK/sco5584*, *glnE/sco2234*, and *glnD/sco5585* and the double mutants *afsQ1/sco4907*&*afsQ2/sco4906* and *amtB/sco5583*&*glnK/sco5584*) and of strains derived from *S. coelicolor* M600 and deleted for genes encoding the global regulator DasR/SCO5231 [8] that controls negatively the expression of proteins involved in the up-take of N-acetylglucosamine (NAG) [9], a component of peptidoglycan, by a phosphotransferase system including the permease NagE2/SCO2907 [10] as well as proteins involved in the intracellular degradation of NAG (NagB/SCO5236, NagK/SCO4285, and NagA/SCO4284) [11].

This study revealed that only 7 deleted strains for the genes *pup*, *glnA2*, *glnA3*, *glnA4*, *dasR*, *glnK*, and *glnE*, to a lesser extent, had a significant positive impact on the total lipid/TAG content. These 7 strains were grown on R2YE either limited (1 mM) or proficient (5 mM) in phosphate (Pi), and their TAG content and level of ACT production were compared to those of the original strain, *S. coelicolor* M145 (*SC*). This study revealed that it was the deletion of the *pup*, *glnA2*, and *dasR* genes that had the strongest positive impact on TAG content in both Pi conditions but more so in Pi proficiency. Unexpectedly, the cardiolipin content of most of these strains (except that of *glnA2* and *pup* mutant strains) was lower than that of the native strain, and that of the *dasR* mutant strain was the lowest, being threefold lower than that of the original strain. Interestingly, ACT production was totally abolished in the *dasR* mutant in both Pi conditions, whereas the deletion of *pup*, *glnA2*, *glnA3*, and *glnA4* was correlated with a significant increase in total ACT production in Pi limitation, when this production was only slightly enhanced in the *pup*, *glnA2*, and *glnA4* mutants and was strongly reduced in the *glnA3* mutant as well as in the *glnE* and *glnK* mutants in Pi proficiency.

Our study confirmed that N limitation is an important trigger of TAG biosynthesis in *SC*, as in most other microorganisms studied [12,13,14,15]. Surprisingly, the high TAG content of 5 of the 7 mutant strains studied (with the exception of *glnA2* and *pup* mutant strains) was correlated with a low cardiolipin content, suggesting that TAG biosynthesis might occur at the expense of that of CL. N limitation also triggers ACT biosynthesis, but the intensity of this trigger was limited by high Pi availability, whereas high Pi availability stimulated TAG biosynthesis. At last, our data indicated that the biosynthesis of TAG and that of ACT were not mutually exclusive, even if many reports in the literature mentioned that a high TAG content was usually correlated with low specialized metabolite production and conversely [16,17,18,19,20]. In conclusion, we propose that the biosynthesis of ACT, which was shown to have anti-oxidant properties [21] and is likely to be induced by oxidative stress (OxS) [22], together with the enhanced or reduced biosynthesis of TAG and CL, respectively, might contribute to the reduction of OxS resulting from nitrogen limitation or from other causes.

## 2. Materials and Methods

### 2.1. Strains and Culture Conditions

The strains used in this study were derived from *S. coelicolor* (*SC*) M145 or M600 [23]. These were generous gifts of Wolfgang Wholleben for strains deleted for genes involved in N metabolism (*glnA/sco2198*, *glnII/sco2210*, *nnar/sco2958*,*gdhA/sco4683*) or its regulation (*glnR/sco4159*, *glnRII/sco2213*, *glnK/sco5584*, *glnE/sco2234*, *glnD/sco5585* and the double mutants *afsQ1/sco4907*&*afsQ2/sco4906* and *amtB/sco5583*&*glnK/sco5584*) as well as in polyamine degradation (GlnA2/SCO2241, GlnA3/SCO6962, GlnA4/SCO1613) [3,4,5,6,7]; of Jean-Luc Pernodet and Hasna Boubakri for the strain deleted for *pup/sco1646* involved in protein degradation [1,2] and of Gilles Van Wezel and Sébastien Rigali for strains derived from *S. coelicolor* M600 including strains deleted for the genes encoding the regulator DasR/SCO5231 [8], the permease NagE2/SCO2907 [10] as well as proteins involved in the intracellular degradation of NAG (NagB/SCO5236, NagK/SCO4285, NagA/SCO4284) [11].

These strains were grown in triplicates on agar plates of the modified R2YE medium [24] devoid of sucrose and with (5 mM final, condition of phosphate proficiency) or without (1 mM final, condition of phosphate limitation) addition of K_2_HPO_4_. This medium is thought to be limited both in nitrogen and phosphate. Its main nitrogen sources are proline (3 g·L^−1^), a mixture of casamino acids Difco (0.1 g·L^−1^) and yeast extract (5 g·L^−1^), whereas free Pi (1 mM as determined by the Pi blue test from Gentaur, France) originates mainly from yeast extract. Approximately 10^6^ spores of the strains were plated on the surface of cellophane disks (Focus Packaging & Design Ltd., Louth, UK) laid down on the surface of solid R2YE plates. The plates were incubated for 48 h or 72 h at 28 °C, in obscurity.

### 2.2. Analysis of the Lipid Content

Approximately 10^6^ spores of the 21 mutant strains and of the native strain of *SC* were plated on cellophane disks deposited on the surface of plates of solid modified R2YE medium proficient in Pi (5 mM). Plates were incubated for 72 h at 28 °C, and mycelial lawns were scraped off the cellophane disks with a spatula and used to determine the global lipid content by Fourier transform infrared spectroscopy (FTIRS, Pike ZnSe ATR-system and FTIR spectrophotometer Vertex 70 from Bruker, Bruker Optics, Germany) as described previously [25]. The accurate determination of the lipid classes present in 7 of these strains (*glnA2/sco2241*, *glnA3/sco6962*, *glnA4/sco1613*, *glnE/sco2234*, *glnK/sco5584*, *pup/sco1646*, *and dasR*/sco5231 mutant strains) was achieved by LC/MS^2^ as described previously [26,27].

### 2.3. Assay of Actinorhodin Production

The 10^6^ spores of the 6 mutant strains and of the native strain of *SC* were plated on cellophane disks deposited on the surface of plates of solid modified R2YE medium, either proficient (5 mM) or limited (1 mM) in Pi. Plates were incubated for 48 h at 28 °C. The mycelium of the strains was scraped off the cellophane disks with a spatula and used to determine the intracellular ACT production, whereas the agar medium present under the cellophane disk was used to determine the extracellular ACT production as described previously [20,24].

## 3. Results

### 3.1. Impact of the Deletion of Genes Encoding Proteins Playing a Role in Nitrogen Metabolism on the Lipid Content of Streptomyces Coelicolor M145

The lipid content of *SC* (M145 and M600) (Appendix A) and of the 21 deletion mutants of genes involved in nitrogen metabolism mentioned in the introduction was first assessed using the Fourier transform infrared spectroscopy (FTIRS) method as described previously [25]. This method gives an indication of the total lipid content of the strains, and results are shown in Appendix A. Six deletion mutants showing a significant variation in their total lipid content in comparison with the original strain, as well as one (*glnE*) that did not, were chosen to determine more accurately their content in different lipid species using LC/MS2 [26], when grown on the solid modified R2YE medium either limited (1 mM) or proficient (5 mM) in phosphate (Pi) at 28 °C for 72 h. Results are shown in Figure 1, Figure 2 and Figure 3.

#### 3.1.1. glnA2, glnA3, and glnA4 Deletion Mutants

*Streptomyces coelicolor* M145 was shown to be able to grow in the presence of high concentrations of polyamines, such as putrescine, cadaverine, spermidine, or spermine, as a sole nitrogen source [3,4,5]. However, these polyamines can also be synthesized intracellularly by bacteria, constituting nitrogen storage molecules that can be degraded in conditions of low external nitrogen availability [7,28]. Genes encoding glutamine synthetase-like proteins were characterized in *SC* and shown to be involved in the degradation of polyamines. These enzymes, GlnA2 (SCO2241), GlnA3 (SCO6962), and GlnA4 (SCO1613), catalyze the glutamylation of their favorite polyamine substrates, which constitutes the first step of their catabolism. GlnA2 possesses the highest specificity towards short-chain polyamines (putrescine and cadaverine) [3], while GlnA3 (SCO6962) prefers long-chain polyamines (spermidine and spermine) [4] and GlnA4 (SCO1613) accepts only monoamines such as ethanolamine [5]. These enzymes thus contribute to the survival of *SC* in conditions of high and potentially toxic external polyamine concentrations [28] or in conditions of low external N availability via the degradation of intracellular polyamines, leading to the re-cycling of N present in these biological molecules.

Analysis of the lipid content of S*. coelicolor* M145 and of these 3 mutant strains grown on solid R2YE medium either limited (1 mM) or proficient (5 mM) in Pi (Figure 1A–C) indicated that (1) the TAG content of *S. coelicolor* M145 did not vary with Pi availability, as reported previously [27]. In contrast, the TAG content of the *glnA2* mutant strain was 2.22 and 2.82, and that of the *gln4* mutant was 1.22- and 1.64-fold higher than that of the original strain in Pi limitation and proficiency, respectively. In these conditions, the TAG content of the *glnA4* mutant was approximately 40% lower than that of the *glnA2* mutant. In contrast, the TAG content of the *glnA3* mutant was similar to that of the original strain in Pi limitation and 1.6-fold higher than that of the latter in Pi proficiency, and its TAG content was also approximately 40% lower than that of the *glnA2* mutant. (2) The phosphatidyl ethanolamine (PE) content of the *glnA2* and *glnA3* mutants was similar to that of the original strain in both Pi conditions, whereas that of the *glnA4* mutant was slightly lower. This was unexpected since GlnA4 was proposed to be the first enzyme of the ethanolamine degradative pathway; its absence was predicted to be correlated with higher ethanolamine and thus PE levels. The absence of this degradative pathway might thus stimulate the expression of alternative ones [29]. (3) Ornithine lipids (OL) were unfortunately co-eluting with PA and Ac-PIM2, however, since OL are phosphate-free lipids known to replace phospholipids (PL) in conditions of Pi limitation [27,30,31,32], their content is likely to be higher in Pi limitation than in Pi proficiency in the original strain. The putative OL content of the original strain and of the *glnA2* mutant were similar, whereas that of the *glnA3* and *glnA4* mutants was over twofold lower than that of the original strain in Pi limitation. However, the co-elution of OL with PA and Ac-PIM2 did not allow rigorous interpretation of these data. (4) The phosphatidylinositol (PI) content was at a similar level in the original strain and in the *glnA2* and *glnA4* mutants but was slightly higher in the *glnA3* mutant in both Pi conditions. In the 3 mutant strains, the PI content was higher in Pi proficiency than in Pi limitation. (5) The cardiolipin (CL) content of the original strain and of the *glnA2* mutant was similar, whereas the CL content of the *glnA3* and *glnA4* mutants was approximately 30 to 40% lower than that of the original strain in both Pi conditions.

**Figure 1 microorganisms-12-01560-f001:**
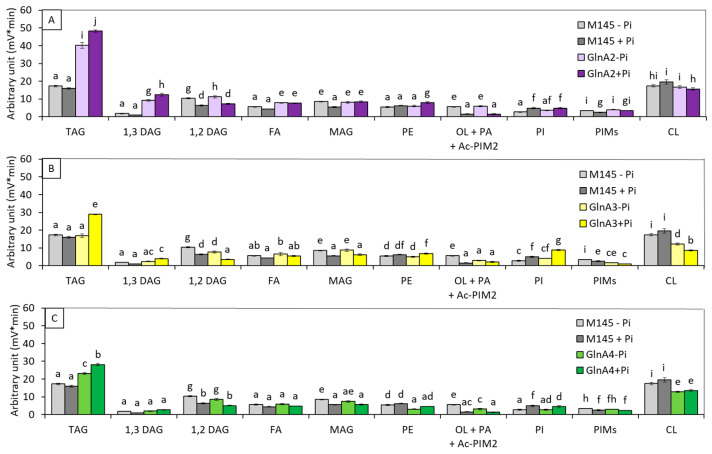
LC/Corona-CAD analysis of the total lipid content of the original strain of *S. coelicolor* M145 (grey histograms) and of derivatives of this strain deleted for *glnA2*/*sco2241* (purple histograms) (**A**), *glnA3*/*sco6962* (yellow histograms) (**B**) or *gnlA4/sco1613* (green histograms) (**C**). The strains were grown on modified solid R2YE medium either limited (1 mM, light color histograms) or proficient (5 mM, dark color histograms) in phosphate at 28 °C for 72 h. TAG, triacylglycerol; DAG, diacylglycerol (1,2 or 1,3); FA, fatty acids; MAG, monoacylglycerol; PE, phosphatidylethanolamine; OL, ornithine lipids; PI, phosphatidylinositol; Ac-PIM2, acetylated phosphatidylinositol mannoside 2; CL, cardiolipid. Mean values are shown as histograms with error bars representing standard error. Means sharing a letter are not significantly different (*p* > 0.05; Tukey-adjusted comparisons).

Our results indicate that a default in internal polyamine degradation leads to nitrogen deficiency that triggers TAG biosynthesis in *SC*, as in most microorganisms studied [12,13,14,15]. Interestingly, external Pi availability did not have any impact on the TAG content of *S. coelicolor* M145 nor on that of *S. lividans* TK24, as reported previously [27], whereas TAG and PI content were higher in Pi proficiency in the *glnA3* mutant strain as well as in the *glnA2* and *glnA4* mutant strains, to a lesser extent. This effect is not clearly understood, but it might be due to the higher expression of these genes in Pi proficiency than in Pi limitation; in consequence, their deletion would have more drastic negative consequences on N availability and thus a positive one on TAG biosynthesis in Pi proficiency than in Pi limitation.

#### 3.1.2. Pup Mutant

Pup/SCO1646 (72 AA) is a prokaryotic ubiquitin-like protein that plays an important role in the turnover and degradation of proteins [1,2]. The covalent attachment of this small protein to specific proteins targets the latter to the 20S proteasome, which achieves their degradation. In the *pup* mutant, protein degradation and thus N turnover is expected to be less efficient than in the original strain leading to reduced N availability that would have a positive impact on TAG biosynthesis.

Analysis of the lipid content of the pup mutant grown in Pi limitation or proficiency (Figure 2) revealed that (1) the TAG content of the *pup* was 1.6- and 2.25-fold higher than that of the original strain in Pi limitation and proficiency, respectively. (2) The phosphatidyl ethanolamine (PE) content of the original strain and of the *pup* mutant were rather similar in both Pi conditions. (3) The putative ornithine lipid (OL) content of the *pup* mutant strain was similar to that of the original strain and, as expected, higher in Pi limitation than in Pi proficiency. (4) The phosphatidylinositol (PI) content of the original strain and of the *pup* mutant strains was similar and higher in Pi proficiency than in Pi limitation. (5) The cardiolipin (CL) contents of the original strain and of the *pup* mutant were similar in both Pi conditions.

**Figure 2 microorganisms-12-01560-f002:**
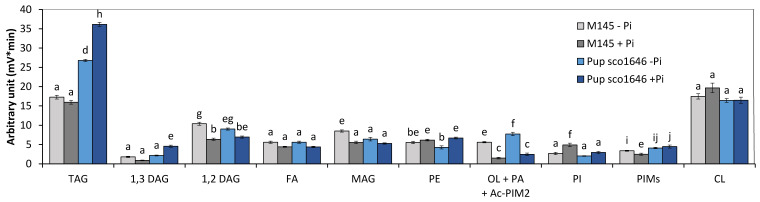
LC/Corona-CAD analysis of the total lipid content of the original strain of *S. coelicolor* M145 (grey histograms) and of derivatives of this strain deleted for *pup*/*sco1646* (marine blue histograms). The strains were grown on modified solid R2YE medium either limited (1 mM, light color histograms) or proficient (5 mM, dark color histograms) in phosphate at 28 °C for 72 h. TAG, triacylglycerol; DAG, diacylglycerol (1,2 or 1,3); FA, fatty acids; MAG, monoacylglycerol; PE, phosphatidylethanolamine; OL, ornithine lipids; PI, phosphatidylinositol; Ac-PIM2, acetylated phosphatidylinositol mannoside 2; CL, cardiolipid. Mean values are shown as histograms with error bars representing standard error. Means sharing a letter are not significantly different (*p* > 0.05; Tukey-adjusted comparisons).

#### 3.1.3. dasR Mutant

The transcriptional regulator DasR was shown to negatively control the uptake of N-acetylglucosamine (NAG), a molecule resulting from the autolytic degradation of cell wall peptidoglycan [8,11], by a phosphotransferase transport system including the permease NagE2/SCO2907 [10]. DasR also negatively controls the intracellular catabolism of NAG by specific enzymes encoded by the genes *nagB/sco5236*, *nagK/sco4285*, and *nagA/sco4284* [11]. However, DasR is a pleiotropic regulator with multiple regulatory targets. It also negatively controls the expression of acetyl-CoA synthetase [33] as well as citrate synthases [34,35], encoding genes that are directly involved in lipid metabolism, as well as that of the regulator *dmdR1*, which negatively controls the expression of genes involved in siderophore biosynthesis and uptake [36,37,38,39]. In consequence, in a DasR mutant, genes involved in the catabolism of NAG and those directly involved in lipid metabolism should be up-regulated, whereas those involved in siderophore-mediated iron uptake should be down-regulated.

Analysis of the total lipid content of the *nagB/sco5236*, *nagK/sco4285*, *nagA/sco4284*, *and nagE2/sco2907* mutant strains by FTIRS (Appendix A) revealed that these strains had a slightly lower lipid content than the original strain, whereas the *dasR* mutant had a twofold higher lipid content than the original strain. Analysis of the lipid content of the *dasR* mutant strain grown in Pi limitation or proficiency in LC/MS2 (Figure 3) revealed that (1) the TAG content of the *dasR* mutant was 1.82- and 1.68-fold higher than that of the original strain in Pi limitation and proficiency, respectively. (2) The phosphatidyl ethanolamine (PE) content of the original strain and of the *dasR* mutant were similar in Pi limitation but the PE content of the *dasR* mutant was 1.5-fold higher than that of the original strain in Pi proficiency. (3) The putative ornithine lipid (OL) contents of the original strain and of the *dasR* mutant were similar and, as expected, higher in Pi limitation than in Pi proficiency. (4) The phosphatidylinositol (PI) content of the original strain and of the *dasR* mutant strain was similar and higher in Pi proficiency than in Pi limitation. (5) The cardiolipin (CL) content of the *dasR* mutant was unexpectedly approximately threefold lower than that of the original strain in both Pi conditions.

**Figure 3 microorganisms-12-01560-f003:**
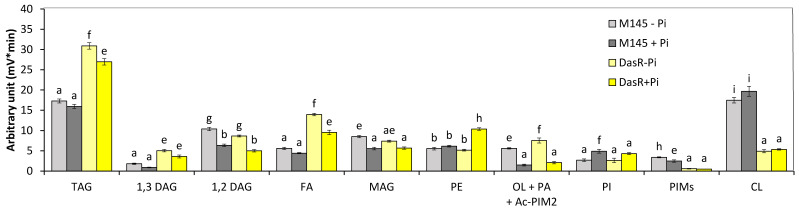
LC/Corona-CAD analysis of the total lipid content of the original strain of *S. coelicolor* M145 (grey histograms) and of derivatives of this strain deleted for *dasR*/sco*5231* (brown histograms). The strains were grown on modified solid R2YE medium either limited (1 mM, light color histograms) or proficient (5 mM, dark color histograms) in phosphate, at 28 °C for 72 h. TAG, triacylglycerol; DAG, diacylglycerol (1,2 or 1,3); FA, fatty acids; MAG, monoacylglycerol; PE, phosphatidylethanolamine; OL, ornithine lipids; PI, phosphatidylinositol; Ac-PIM2, acetylated phosphatidylinositol mannoside 2; CL, cardiolipid. Mean values are shown as histograms with error bars representing standard error. Means sharing a letter are not significantly different (*p* > 0.05; Tukey-adjusted comparisons).

#### 3.1.4. glnK and glnE Deletion Mutants

The *glnK*/*sco5584* gene of *S. coelicolor* M145 encodes a P-II-like nitrogen regulatory protein. Regulatory PII proteins are thought to sense and integrate the intracellular levels of carbon (2-oxogluratate) and nitrogen (glutamine), as well as the energetic status (ATP/ADP ratio), and control the activities of various transcription factors, transport systems, or enzymes by direct interaction with the latter [40,41,42,43].

The *glnE*/*sco2234* gene of *S. coelicolor* M145 encodes an adenylyltransferase that adenylates glutamine synthase I/GlnA/SCO2198 [44]. GlnA plays a crucial role in nitrogen assimilation since it catalyzes the condensation of glutamate and ammonium to yield glutamine. The adenylylation of GlnA by GlnE downregulates GlnA activity after an ammonium shock in order to maintain glutamate/glutamine homeostasis [45]. In contrast to the *E. coli* situation, in *Streptomyces*, neither GlnK nor GlnD are required for the regulation of GlnE in response to nitrogen availability [46].

Analysis of the lipid content of these 2 mutant strains grown in Pi limitation or proficiency (Figure 4A,B) revealed that (1) the TAG content of the *glnE* mutant was rather similar to that of the original strain (1.1- and 1.2-fold higher in Pi limitation and Pi proficiency, respectively), whereas that of the *glnK* mutant was 1.23- and 2-fold higher than that of the original strain in Pi limitation and Pi proficiency, respectively. (2) The phosphatidyl ethanolamine (PE) content of the original strain and of the *glnK* mutant were similar in both Pi conditions, whereas that of the *glnE* mutant was approximately twofold lower than that of the original strain in Pi limitation. (3) The putative ornithine lipid (OL) contents of the original strain and of the *glnE* mutant were similar in both Pi conditions, whereas that of the *glnK* mutant strains was lower than that of the original strain in both Pi conditions. (4) The phosphatidylinositol (PI) content of the original strain and of the *glnK* mutant were similar in both Pi conditions, whereas that of the *glnE* mutant was approximately 1.5- to 2-fold higher than that of the original strain in Pi limitation and proficiency, respectively. (5) The cardiolipin (CL) content of the *glnK* mutant and the *glnE* mutant, to a lesser extent, was lower than that of the original strain in both Pi conditions and were lower in Pi proficiency than in Pi limitation.

The deletion of *glnE* led only to limited TAG accumulation in both Pi conditions, whereas the deletion of *glnK* led to twofold increase in TAG content compared to the original strain, but mainly in Pi proficiency. The deletion of genes encoding PII proteins was also reported to enhance fatty acid biosynthesis in *E. coli* [47]. This might be related to the negative impact that these proteins exert on the activity of acetylCoA-carboxylase as reported in *Synechococcus* [48]. Furthermore, the activity of GlnK is stimulated when the ATP/ADP ratio is high [40], as in Pi proficiency; the deletion of *glnK* might have more drastic consequences in Pi proficiency than in Pi limitation. At last, interestingly, the concomitant deletion of *amtB/sco5583* and *glnK/sco5584* (Appendix A) did not lead to TAG accumulation, suggesting that the ammonium transporter AmtB might be involved in the efflux of intracellular ammonium [49] resulting from the usual degradation of proteins, amino acids, and polyamines. Its deletion would thus lead to an increase in intracellular N availability that would impair the triggering of TAG biosynthesis.

### 3.2. Impact of the Deletion of Genes Encoding Proteins Playing a Role in the Degradation of Nitrogen-Containing Biological Molecules on the Production of Actinorhodin in Streptomyces Coelicolor M145

Since N as well as P limitations are known to trigger specialized metabolite biosynthesis in *Streptomyces* species [50,51], we assessed intra- and extracellular production of the blue polyketide actinorhodin (ACT) in our mutants. This study revealed that ACT production was totally abolished in the *dasR* mutant in both Pi conditions and thus could not be assayed. The biosynthesis of other specialized metabolites was also reported to be abolished in *dasR* mutants of other *Streptomyces* species [52,53], as well as in *Saccharopolyspora erythraea* [54]. Interestingly, the original strain and the 6 mutant strains excreted the most ACT produced when grown in conditions of Pi limitation (Figure 5A), whereas ACT remained mainly intracellular when the strains were grown in conditions of Pi proficiency (Figure 5B).

In Pi limitation, the level of total ACT production of the *glnA2*, *glnA3*, and *glnA4* mutant strains, as well as the *pup* mutant, was 1.4- to 1.6-fold higher than that of the original strain. In these conditions, the *glnA2*, *pup*, and *gln4* mutants had a 2.21-, 1.46-, and 1.19-fold higher TAG content than that of the wild-type strain, respectively, whereas GlnA3 had a TAG content similar to that of the original strain. The level of ACT production of the *glnK* and *glnE* mutants was also similar to that of the original strain, and their TAG was only slightly higher than that of the original strain. 

In Pi proficiency, the *glnA2*, *glnA4* and *pup* mutant strains are producing 1.1- to 1.3-fold more intracellular ACT than the original strain. In these conditions, the *glnA2*, *glnA3*, *glnA4*, and *pup* had a 3-, 1.8-, 1.75-, and 2.25-fold higher TAG content than the original strain, respectively. In contrast, the *glnE*, *glnA3*, and *glnK* mutants produces 12-, 6-, and 3.4-fold less ACT than the original strain. In these conditions, the *glnE*, *glnA3*, and *glnK* mutants had a 1.3-, 1.81-, and 2.1-fold higher TAG content than the original strain.

## 4. Discussion

In this issue, we demonstrated that mutant strains deleted for the genes *pup* and *glnA2* as well as for *glnA4* and *glnA3*, to a lesser extent, that encode proteins involved in the degradation of N-containing biological molecules, proteins, and polyamines, accumulate a higher level of TAG than the original strain. Since TAG biosynthesis is known to be triggered in conditions of N deprivation in most microorganisms [12,13,14,15], the higher TAG content of the *pup* and *glnA2* mutant strains, compared to that of *glnA3* and *glnA4* mutant strains, suggested that the degradation of the Pup-targeted proteins by the proteasome and of short-chain polyamines (putrescine and cadaverine) by GlnA2 play a more important role in the internal N supply via the recycling of N present in these biological molecules than GlnA3 and GlnA4 involved in the degradation of long-chain polyamines (spermidine and spermine) and ethanolamine, respectively. This might be due to the higher intracellular abundance of the GlnA2 substrates than of the GlnA3 and GlnA4 substrates. As a result, the deletion of *pup* and *glnA2* led to more severe nitrogen limitation, resulting in higher TAG content than that of *glnA3* and *glnA4.*

The deletion of *dasR* also had a strong positive impact on TAG biosynthesis, whereas the deletion of genes belonging to the DasR regulon involved in NAG uptake and degradation did not (Appendix A). Since DasR is a pleiotropic regulator with numerous regulatory targets [55], its high TAG content is likely to result from multiple causes. TAG accumulation in the DasR mutant cannot be due to the negative control it exerts on the expression of enzymes involved in NAG uptake and degradation, since in the *dasR* mutant, the expression of these enzymes is higher than in the original strain [8,11] and should result in an enhanced N availability that has a negative impact on TAG biosynthesis. The high TAG content of the *dasR* mutant might thus be due to the negative regulation that DasR exerts on the expression of genes encoding citrate synthases, enzymes catalyzing the conversion of citrate into acetylCoA [34,35] and acetyl-CoA synthetase [33]. These regulatory features were demonstrated in *Saccharopolyspora erythraea*, but not in *SC* yet. However, if these enzymes are over-expressed in the *dasR* mutant of *SC*, as they are in the *dasR* mutant *S. erythraea*, an excess acetylCoA might be generated and used for the biosynthesis of fatty acids present in TAG. Furthermore, DasR was shown to repress the expression of the regulator *dmdR1*, which negatively controls the expression of genes involved in siderophore biosynthesis and uptake in *SC* [36,37,38,39]. In consequence, in a DasR mutant, these genes are not expressed, and siderophore-mediated iron uptake does not occur resulting into iron deprivation. Interestingly, iron deprivation was shown to promote TAG accumulation, at least in *Chlamydomonas species* [56,57]. Since iron is a necessary co-factor of numerous enzymes, including enzymes of the TCA cycle (aconitase, citrate synthase, isocitrate dehydrogenase, and succinate dehydrogenase) and of the respiratory chain, reduced iron availability ought to result in low TCA activity, whereas TCA activity is crucial for N assimilation. Iron deprivation might thus indirectly result in low N assimilation, triggering TAG biosynthesis. Interestingly, the lower TAG content of the d*asR* mutant in Pi proficiency than in Pi limitation might be due to the fact that Pi could be transported as an iron chelate [58] and thus co –Pi/iron transport might reduce the severity of iron shortage and thus TAG accumulation.

At last, we noticed that the *dasR* mutant had the lowest cardiolipin (CL) content of all strains. Its CL content was approximately threefold lower than that of the original strain in both Pi conditions. The very low CL content of the *dasR* mutant was unexpected and is not understood, but an anti-correlation between CL and TAG content was previously reported in *Saccharomyces cerevisiae* [59], suggesting that these two lipid species could be inter-converted. DasR might positively control CL biosynthesis as it does for iron uptake. CL and iron are both known to play an important role in the good functioning of enzymes in the respiratory chain [60,61,62], so the co-regulation of these two processes by DasR could make sense. However, *sco1389*, encoding a putative eukaryotic-like cardiolipin synthase [30,63], was not found among the DasR target genes [55]. In contrast, *sco5773*, encoding putative phosphatidylglycerol phosphate synthase [30,63], could possibly be one of the DasR target genes since it is located downstream, transcribed in the same direction, and perhaps co-transcribed with *sco5751*, which is listed as a potential DasR target gene in Appendix A of [55]. If, as in other organisms, CL plays an important role in the good functioning of enzymes of the respiratory chain [60,61,62] and thus of respiration, the low CL content of the *dasR* mutant might lead to a reduced respiration and thus a lower generation of OxS, which was proposed to be an important trigger of ACT biosynthesis [22]. Indeed, the deletion of *dasR* totally abolished ACT production in both Pi conditions, whereas the deletion of *glnA2*, *glnA3*, *glnA4*, and *pup* led to an increase in ACT production (1.5-fold in average), compared to the original strain, in Pi limitation. As expected, in Pi proficiency, this increase was more moderate (1.2 in average) in the *glnA2*, *glnA4*, and *pup* mutant, since ACT biosynthesis is known to be repressed in this condition [50,64]. In contrast and unexpectedly, ACT production was sixfold lower in the *glnA3* mutant than in the original strain in Pi proficiency. The deletion of *glnE* and *glnK* also led to a strong reduction of ACT production (12- and 3.4-fold, respectively), but only in Pi proficiency. Interestingly, in all strains, ACT was mainly excreted in Pi limitation, whereas in Pi proficiency, ACT remained intracellular. Since we have previously demonstrated that ACT bears an anti-oxidant function [21] and is thus likely to be triggered by oxidative stress [65,66,67], the strains producing high levels of ACT are likely to be those experiencing high levels of oxidative stress, and conversely. The *glnA2*, *glnA3*, *glnA4*, and *pup* deletion mutants that are suffering from N limitation are likely to suffer from oxidative stress. Indeed, some reports in the literature mention that N limitation induces oxidative stress (OxS) [68] and that OxS plays a positive role in the triggering of lipid/TAG biosynthesis in various organisms [48,69,70]. The total absence of ACT biosynthesis in the *dasR* mutant might be attributed to a low level of OxS due to the important storage of acetylCoA as TAG that limits the activation of the oxidative metabolism, the generator of OxS, as well as to iron deficiency that limits the generation of OxS by the Fenton reaction [71]. The very low ACT production of the *glnE*, *glnK*, and *glnA3* mutant strains in Pi proficiency suggested that oxidative stress was lower in these mutants than in the original strain. GlnE is known to inhibit GlnA activity [44], so in its absence, N assimilation might be stimulated, resulting in N proficiency that is correlated with a lower level of OxS and thus lower ACT biosynthesis. The lower OxS of the *glnK* mutant might also result from better N assimilation. At last, the low level of OxS of the GlnA3 mutant, which is correlated with reduced ACT biosynthesis, might be due to its high spermine and spermidine content. Indeed, several reports in the literature mention that these molecules protect the cell against OxS [72,73].

In conclusion, our study confirmed that N limitation is an important trigger of TAG biosynthesis in *SC*, as in most microorganisms [12,13,14,15]. Surprisingly, the high TAG content of 5 of the 7 mutant strains studied (with the exception of *glnA2* and *pup* mutant strains) was correlated with a low cardiolipin content, suggesting that TAG biosynthesis might occur at the expense of that of CL. N limitation also triggers specialized metabolite biosynthesis, but the intensity of this trigger was limited by high Pi availability, whereas high Pi availability stimulated TAG biosynthesis. At last, our data indicated that the biosynthesis of TAG and that of ACT were not mutually exclusive, even if many reports in the literature mentioned that a high TAG content was usually correlated with low specialized metabolite production and conversely [16,17,18,19,20]. Interestingly, the enhanced and reduced TAG and CL content, respectively, and the triggering of ACT biosynthesis might all contribute to the lowering of OxS. Indeed, the storage of acetylCoA as TAG limits the feeding of the TCA cycle and thus the activation of the oxidative metabolism that generates OxS, whereas low CL content might contribute to a lower respiratory activity that would lead to a reduction of OxS. At last, ACT, via its anti-oxidant activity [21], would also limit OxS resulting from nitrogen limitation or from other causes.

## Figures and Tables

**Figure 4 microorganisms-12-01560-f004:**
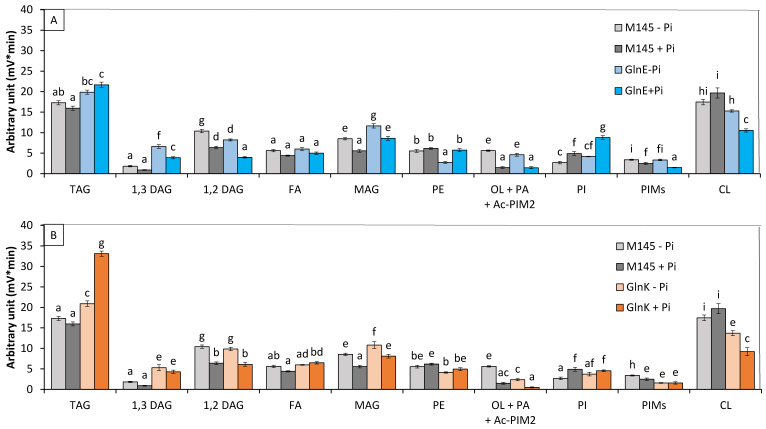
LC/Corona-CAD analysis of the total lipid content of the original strain of *S. coelicolor* M145 (grey histograms) and of derivatives of this strain deleted for *glnE*/*sco2234* (blue histograms) (**A**) and *glnK*/*sco5584* (orange histograms) (**B**). The strains were grown on modified solid R2YE medium either limited (1 mM, light color histograms) or proficient (5 mM, dark color histograms) in phosphate, at 28 °C for 72 h. TAG, triacylglycerol; DAG, diacylglycerol (1,2 or 1,3); FA, fatty acids; MAG, monoacylglycerol; PE, phosphatidylethanolamine; OL, ornithine lipids; PI, phosphatidylinositol; Ac-PIM2, acetylated phosphatidylinositol mannoside 2; CL, cardiolipid. Mean values are shown as histograms with error bars representing standard error. Means sharing a letter are not significantly different (*p* > 0.05; Tukey-adjusted comparisons).

**Figure 5 microorganisms-12-01560-f005:**
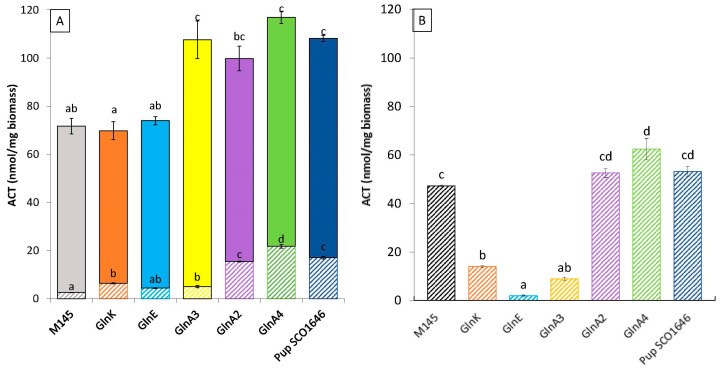
Quantification of total cellular actinorhodin produced by the original strain of *S. coelicolor* M145 (grey histograms) and derivatives of this strain deleted for the genes *glnK*/*sco5584* (orange histograms) and *glnE*/*sco2234* (blue histograms), *glnA3*/*sco6962* (yellow histograms), *glnA2*/*sco2241* (purple histograms), *gnlA4/sco1613* (green histograms) and *pup/sco1646* (marine blue histograms) on modified solid R2YE medium either limited (1 mM) (**A**) or proficient (5 mM) in phosphate (**B**) grown at 28 °C for 72 h. Plain and hatched parts of the histograms represent extracellular and intracellular ACT production, respectively. Mean values are shown as histograms with error bars representing standard error. Means sharing a letter are not significantly different (*p* > 0.05; Tukey-adjusted comparisons).

## Data Availability

The data supporting the findings of this study are available from the corresponding author.

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
