# Peer review of "Impact of the Deletion of Genes of the Nitrogen Metabolism on Triacylglycerol, Cardiolipin and Actinorhodin Biosynthesis in Streptomyces coelicolor"

_microorganisms, 2024, doi:10.3390/microorganisms12081560_

Round 1
Reviewer 1 Report
Comments and Suggestions for Authors
Review
for the article entitled “Impact of the deletion of genes of the nitrogen metabolism on triacylglycerol, cardiolipin and actinorhodin biosynthesis in Streptomyces coelicolor.”
The article is consistent with the subject matter of the journal “Microorganisms” and contains new information regarding the impact of the deletion of genes involved in nitrogen metabolism on triacylglycerol, cardiolipin and actinorhodin biosynthesis in Streptomyces coelicolor. It is therefore recommended for publication in the journal. However, the authors are encouraged to undertake a comprehensive revision of the manuscript in order to address the following areas for improvement:.
Point 1: Remove the dot at the end of the article title.
Point 2: Page 1, line 6 – Please verify the following sentence: “And Sonia Abreu et Clara Lejeune contributed equally to this work.”
Point 3: The abstract should not exceed 200 words.
Point 4: In the text, reference numbers should be placed in square brackets [ ], for example [1], [1–3] or [1,3].
Point5: page 2, It is recommended to modify the introduction. Focus on the current state of the research area and provide references to key publications related to the genetic basis of nitrogen metabolism on secondary metabolite synthesis in Streptomyces coelicolor. The introduction should clearly define the purpose and novelty of the work. The main conclusions of the work should be concise. The conclusions presented in the last three sections (lines 51-93) are cumbersome and repeated in other sections of the paper.
Point 6: page 3, lines 116-122. In addition to the reference, give a detailed description of the conditions for determining the lipid classes, actinorhodin.
Point 7: page 3: It is recommended that the statistical analysis of the data be included in the “Materials and Methods” section.
Author Response
Point 1: Remove the dot at the end of the article title.
This was done
Point 2: Page 1, line 6 – Please verify the following sentence: “And Sonia Abreu et Clara Lejeune contributed equally to this work.”
Thank you for this comment "And" was canceled
Point 3: The abstract should not exceed 200 words.
The abstract is now shorter.
Point 4: In the text, reference numbers should be placed in square brackets [ ], for example [1], [1–3] or [1,3].
This was corrected
Point5: page 2, It is recommended to modify the introduction. Focus on the current state of the research area and provide references to key publications related to the genetic basis of nitrogen metabolism on secondary metabolite synthesis in Streptomyces coelicolor. The introduction should clearly define the purpose and novelty of the work. The main conclusions of the work should be concise. The conclusions presented in the last three sections (lines 51-93) are cumbersome and repeated in other sections of the paper.
In France that is summer holidays so I am on holidays and I cannot adress this comment right now but I will try to adress it when I come back at work at the end of august.
Point 6: page 3, lines 116-122. In addition to the reference, give a detailed description of the conditions for determining the lipid classes, actinorhodin.
I have providing some missing details.
Point 7: page 3: It is recommended that the statistical analysis of the data be included in the “Materials and Methods” section.
The person in charge of the statistical analysis is on holiday and I cannot contact her before september.
Reviewer 2 Report
Comments and Suggestions for Authors
This is an interesting study that explores the impact of genes involved in nitrogen metabolism on the production of triacylglycerol, cardiolipin, and actinorhodin in Streptomyces coelicolor. The rationale is clear, the introduction provides sufficient information to place in context the study and the results are relevant for the specialized community. Three aspects need attention, to improve the manuscript:
-The methodology is concise and does not provide enough details to allow reproducibility of results. The manuscript is short, so expanding this section will not make the manuscript lengthy.
- The methodology section needs a subsection that includes the statistical analysis applied to data.
-I suggest changing the way the statistical analysis is shown in the figures. The authors stated in Figure 1: "Means sharing a letter are not significantly different". However, if you see bars with the letter "a" across panel C, the bars are likely different in high but should be taken as not significantly different. This makes no sense.
Author Response
Comments 1: The methodology is concise and does not provide enough details to allow reproducibility of results. The manuscript is short, so expanding this section will not make the manuscript lengthy.
I have added some details in the material and methods section in the new version of the manuscript.
Comments 2: The methodology section needs a subsection that includes the statistical analysis applied to data.
The person in charge of the statistical analysis is currently on holidays I cannot contact her before september.
Comments 3: I suggest changing the way the statistical analysis is shown in the figures. The authors stated in Figure 1: "Means sharing a letter are not significantly different". However, if you see bars with the letter "a" across panel C, the bars are likely different in high but should be taken as not significantly different. This makes no sense.
The person in charge of the statistical analysis is currently on holidays I cannot contact her before september.
Reviewer 3 Report
Comments and Suggestions for Authors
The primary research question addressed in this study is how the deletion of genes involved in nitrogen metabolism affects the biosynthesis of triacylglycerol (TAG), cardiolipin (CL), and actinorhodin (ACT) in Streptomyces coelicolor. The study is original in its approach to understanding the interplay between nitrogen metabolism and lipid metabolism in Streptomyces coelicolor. It addresses a gap in the field by exploring the specific impact of gene deletions on TAG and CL biosynthesis, which are crucial for understanding the metabolic pathways and potential industrial applications of this bacterium. The exploration of the role of nitrogen limitation in the production of secondary metabolites like ACT is also a significant contribution. Compared to other published material, this paper adds to the understanding of how nitrogen availability and metabolism regulate lipid and secondary metabolite production in Streptomyces species. The detailed analysis of the lipid content and ACT production in strains with specific gene deletions provides new insights into the regulatory mechanisms of these processes. The findings challenge the general assumption that high TAG content is usually correlated with low specialized metabolite production, offering a more nuanced view of the metabolic interplay. The conclusions drawn by the authors are generally consistent with the evidence and arguments presented. The study demonstrates a clear impact of nitrogen metabolism gene deletions on TAG and CL biosynthesis and ACT production. The references cited in the paper are appropriate and relevant to the study.
Overall, the manuscript presents a valuable contribution to the field, offering new insights into the complex interplay between nitrogen metabolism and lipid biosynthesis in Streptomyces coelicolor. With some enhancements in methodology and a more detailed discussion of the implications, this work could significantly advance our understanding of microbial metabolism.
Major concerns
1. Incorporating transcriptomic or proteomic analysis could help elucidate the molecular mechanisms underlying the observed metabolic changes.
2. About Figure Legends. Ensuring that all figures have detailed and informative legends would aid in the understanding of the data presented.
3. Statistical Analysis. Providing more detailed statistical analysis, such as p-values or confidence intervals, would strengthen the claims made based on the data.
4. Why choose Streptomyces coelicolor? Is there anything special about the N metabolism of Streptomyces coelicolor?
Author Response
Major concerns
Comment 1 Incorporating transcriptomic or proteomic analysis could help elucidate the molecular mechanisms underlying the observed metabolic changes.
Yes of course but such analyses are beyond the scope of this article.
Comment 2: About Figure Legends. Ensuring that all figures have detailed and informative legends would aid in the understanding of the data presented.
We had the feeling that our figures legends were correct. Would it be possible for the reviewer to say more clearly what is missing?
Comments 3: Statistical Analysis. Providing more detailed statistical analysis, such as p-values or confidence intervals, would strengthen the claims made based on the data.
We state in the figure legend "Means values are shown as histograms with error bars representing standard error. Means sharing a letter are not significantly different (P > 0.05; Tukey-adjusted comparisons)." Would it be possible for the reviewer to say more precisely what he/she is requesting?
Comment 4: Why choose Streptomyces coelicolor? Is there anything special about the N metabolism of Streptomyces coelicolor?
Streptomyces coelicolor is the historical model strain in our field to study regulation/dysregulation of antibiotics biosynthesis since it is a strong antibiotic producer. We also published some work demonstrating that S. coelicolor has indeed an altered nitrogen metabolism (doi: 10.1016/j.resmic.2023.104177 and doi.org/10.3390/ijms232314792)